# You Cannot Hit Snooze on OSA: Sequelae of Pediatric Obstructive Sleep Apnea

**DOI:** 10.3390/children9020261

**Published:** 2022-02-15

**Authors:** Selena Thomas, Shefali Patel, Prabhavathi Gummalla, Mary Anne Tablizo, Catherine Kier

**Affiliations:** 1Department of Pediatrics, Stony Brook University Medical Center, Stony Brook, NY 11794, USA; selenaalexthomas@gmail.com; 2Department of Pediatrics, Valley Children’s Hospital, Madera, CA 93636, USA; pshefali20@gmail.com (S.P.); mtablizomd@gmail.com (M.A.T.); 3Department of Pediatric Sleep Medicine, Valley Hospital, Ridgewood, NJ 07450, USA; prabhavathi.gummalla@gmail.com; 4Department of Pediatrics, Stanford University School of Medicine, Palo Alto, CA 94304, USA

**Keywords:** pediatric obstructive sleep apnea, obstructive sleep apnea sequelae, endothelial dysfunction, cardiovascular disease, metabolic dysfunction, neurocognitive impairment, psychosomatic syndromes, intermediate mechanisms, continuous positive airway pressure, sleep-disordered breathing

## Abstract

Pediatric obstructive sleep apnea (OSA) has been shown to not only affect the quality of sleep, but also overall health in general. Untreated or inadequately treated OSA can lead to long-term sequelae involving cardiovascular, endothelial, metabolic, endocrine, neurocognitive, and psychological consequences. The physiological effects of pediatric OSA eventually become pathological. As the complex effects of pediatric OSA are discovered, they must be identified early so that healthcare providers can be better equipped to treat and even prevent them. Ultimately, adequate management of OSA improves overall quality of life.

## 1. Introduction

Obstructive sleep apnea (OSA), part of the spectrum of sleep-disordered breathing (SDB), is characterized by frequent arousals, apneas, and hypopneas, and can be associated with reduction in blood oxygen saturation and hypoventilation during sleep in children [1]. The prevalence of children with OSA is around 1–5.8% but is rising due to the obesity pandemic [2]. First-line treatment of pediatric OSA is an adenotonsillectomy (AT), which directly addresses adenoid and tonsillar hypertrophy associated with pediatric OSA [3]. The success rates of adenotonsillectomy vary depending on the population studied but residual OSA rates can be as high as 40 to 75% in children [1]. For children who have OSA that persists into later childhood, alternate forms of treatment and management are necessary to maintain quality of life. Continuous positive airway pressure (CPAP) may be necessary for a select group of pediatric patients with OSA. Other options need to be explored and studied to prevent long term sequalae.

Pediatric obstructive sleep apnea (OSA) has adverse effects as a result of disruption of sleep and abnormal ventilation [4]. Behavioral and learning issues are commonly seen in younger children, which can present as attention problems, hyperactivity, irritability, and poor school performance. Long term, if untreated, pediatric OSA can lead to adverse cardiovascular, endothelial, metabolic, endocrine, neurocognitive, and psychological outcomes that affect quality of life [5]. The costs of treating long-term sequelae pose an additional burden on children, their families, as well as the entire health care system. By increasing awareness of the potential long-term sequelae of OSA and by adequate screening of children, OSA can be identified and treated accordingly. Ideally, early identification and treatment could potentially affect the trajectory of the disease in children. Figure 1 shows a schematic of the sequelae that will be detailed in this review.

## 2. Cardiovascular Diseases and OSA

Untreated or inadequately treated OSA in children can lead to systemic and pulmonary hypertension, postural orthostatic hypertension syndrome, cardiac arrhythmias, coronary artery changes, and cerebrovascular changes [6].

### 2.1. Hypertension

Childhood OSA is an independent risk factor for adverse blood pressure outcomes [7]. Hypertension in pediatric OSA is due to multifactorial pathogenesis [8]. Recurrent hypoxemia and hypercapnia in pediatric OSA lead to activation of the sympathetic nervous system with consequent increases in catecholamine levels [7], which persist during the daytime and contribute to the development of hypertension.

Intermittent elevations in systemic blood pressure during sleep have been observed in children with OSA [9]. Children with moderate to severe OSA were found to have higher nocturnal systolic blood pressure and reduced nocturnal drop of systolic blood pressure at 10-year follow up [10]. Mean blood pressure variability was noted to be higher during wakefulness and during sleep in children with OSA compared to children with primary snoring [10]. The nocturnal dipping of the mean blood pressure was smaller in children with OSA compared to those with primary snoring [9].

Pulmonary hypertension has been reported in children with OSA, although it is rare in noncardiac patients. The overall prevalence of pulmonary hypertension in pediatric OSA patients is low (1.8%) [9].

### 2.2. Cardiac Arrhythmias

There have been limited studies on the relationship between OSAS and cardiac arrhythmia in children. Cardiac arrhythmias during sleep include nonsustained ventricular tachycardia (VT), sinus pause or arrest, second degree atrioventricular (AV) conduction block, and frequent premature ventricular contractions (PVCs > 2 bpm). Although cardiac arrhythmias may be seen in pediatric OSA and may be an incidental finding during an overnight polysomnography for a child being assessed for snoring or sleep apnea, the true prevalence and clinical significance of cardiac arrhythmias in pediatric OSA has not been well established. A retrospective study of 124 children diagnosed with OSA (after excluding patients with airway, cardiac, congenital, or genetic abnormalities), with a mean age of 6.7 ± years, was assessed for the presence of arrhythmias and heart rate variability. The cases were categorized based on the severity of OSA as mild, moderate, or severe. All cases demonstrated sinus arrhythmias (sinus rhythm with varying R-R intervals), but only 2.4% (only 3 cases) had occasional PAC. There were no other arrhythmias, such as PVC, AV block, VT, or SVT. Most cases (80.7%, 100 of 124 cases) had an episode of sinus bradycardia (with heart rate less than normal for age). This study showed that the incidence of arrhythmias in pediatric OSA was low and benign arrhythmias were reported. Sinus arrhythmia is a normal variation found even in healthy children. Sinus bradycardia was common but showed no clinical significance in this study. Standard deviation of heart rate (SD-HR), minimum heart rate (min-HR), and maximum heart rate (max-HR) values in NREM and REM stages were assessed. In NREM sleep, there were no significant differences among the mild, moderate, and severe OSA groups. In REM, however, the SD-HR and max-HR values were significantly higher in the severe OSA group. These data may be indirect evidence of increased heart rate variability in severe OSA [11]. Another retrospective cross-sectional study of children below 15 years of age with sleep-disordered breathing (SDB) suggested that heart rate variability was significantly increased in sleep in those with severe OSA compared to those with mild OSA [9].

QT dispersion and P-wave dispersion have also been studied in pediatric OSA. QT dispersion may be associated with a higher risk for ventricular arrhythmia and has been reported in children with severe OSA [9]. P-wave dispersion was found more in children with severe OSA in a study of 44 children at 1–12 years of age [12]. This implies that cardiac arrhythmias occur along a continuum and can be variable and benign except may be in severe OSA. More studies are required to determine links about the pathogenesis of arrhythmias in OSA. 

### 2.3. Coronary Artery Changes and Atherogenesis

Elevated plasma P-selectin levels, a marker of platelet activation, was noted in children with OSA [13]. The activation of P-selectin is mediated by inflammatory processes linked to atherogenesis [13]. Additionally, hypoxemia and sleep fragmentation secondary to OSA can promote both sympathetic activity and reaction oxygen species (ROS) formation, which further cause platelet activation and upregulation of adhesion molecules, respectively. These data support the postulation that pediatric OSA can promote the onset of atherosclerosis and subsequent CAD.

### 2.4. Cerebrovascular Abnormalities

OSA is implicated in neuropsychological deficits and neuronal brain injury. The underlying mechanisms include reduced cerebrovascular perfusion, blood gas abnormalities, and neuronal injury secondary to long-term oxygen saturation abnormalities [14]. Children are especially vulnerable to changes in cerebral blood flow (CBF) compared to adults due to having a higher metabolic rate, a higher resting CBF, and a narrower range of autoregulation [15,16]. This makes children particularly vulnerable to brain injury during critical periods [17].

Hypoxia-induced lactate production and ROS formation can also leave brain regions vulnerable to neuronal injury. Increased prevalence of OSA was noted amongst children with arterial ischemic strokes (AIS), implying an association between OSA and cerebrovascular stroke [18]. This finding highlights the issue that even children may possibly be vulnerable to very significant sequelae as adults. Better screening and prompt identification and treatment of OSA can prevent such morbidity.

## 3. Endothelial Dysfunction

The endothelium is a thin layer of cells lining the blood vessels. It releases substances that control the vascular tone, blood clotting, immune function, and platelet activation. Endothelial dysfunction secondary to OSA can cause adverse effects on various organ systems. Various studies have documented endothelial dysfunction as an early marker for cardiovascular disease, and endothelial dysfunction has been noted as a potential outcome of pediatric OSA [19]. It is assessed with a modified hyperemic test by cuff-induced occlusion of the brachial artery. In nonobese children ages 6–9 years with OSA, significant increases in the time to attain peak reperfusion (Tmax) were noted. Endothelial dysfunction was worse in children with pediatric OSA than in children with primary snoring. Additionally, it was noted that frequent arousals due to airway obstruction in sleep may be an independent risk factor for endothelial dysfunction [19]. Further research is needed to address the comorbid mitigating factors such as obesity, family history, ethnicity etc.

### 3.1. Inflammatory and Immune Markers

The mechanisms that lead to endothelial dysfunction are multifactorial. Intermittent hypoxia during sleep in children with OSA leads to systemic low-grade inflammatory processes with worsening oxidative stress, leading to interactions of endothelial, platelet, and inflammatory cells [20]. This leads to an enhanced proatherogenic state and upregulation of adhesion molecules and other markers of inflammation [14,21,22]. During hypoxia, there is increased expression of the transition factors, hypoxia-induced factor I, inducible factor 1, stromal cell-derived factor-1 (SDF-1), and vascular endothelial growth factor (VEGF) from the endothelial cells [21]. It has been noted that the numbers of circulating endothelial progenitor cells (EPCs) are increased in children with OSA. In children with OSA, serum VEGF levels were significantly higher in polysomnographic-confirmed OSA when compared to those with mild to no disease. Additionally, positive correlations were noted between VEGF concentrations, respiratory disturbance index, and O2 saturations of less than 90% during sleep [22]. Increased leukocyte production in children with OSA is also implicated in the development of atherosclerosis secondary to endothelial damage [23]. Endothelial dysfunction of OSA patients has also recently been associated in subjects with vasomotor rhinitis, where vascular hyperreactivity to chemical or thermal stimuli has been correlated with the severity of obstructive apneas [24,25]. The proposed mechanism is related to circulating cytokines and inflammatory markers released secondary to nasal congestion and eosinophilic inflammation, which may increase the risk of OSA and subsequent endothelial dysfunction. These findings raise the question about the onset and trajectory of atopic disease in children with respect to OSA. Further studies are needed in pediatric populations [24]. 

### 3.2. Renin–Angiotensin–Aldosterone System

The renin–angiotensin–aldosterone system (RAAS) is also implicated in the association of OSA with endothelial dysfunction. During OSA, when intermittent hypoxia occurs, RAAS is activated, leading to elevated blood pressure in sleep. Additionally, patients with OSA were found to have increased levels of circulating angiotensin II and aldosterone [26], which contribute to vasoconstriction [26]. This was documented in both adult and pediatric populations [26]. This may be the other associated mechanistic reason for the reported hypertension in children with OSA.

### 3.3. Leptin

Children with OSA are at higher risk of developing obesity and endothelial dysfunction [22]. OSA can lead to increased leptin levels and reduced leptin sensitivity, leading to overeating in children [27]. Leptin receptors are also present in the endothelium. Therefore, the increase in leptin in children with OSA can give rise to endothelial damage via reactive oxygen species (ROS) and atherosclerotic changes [28]. This implies that the systemic effects of OSA is multifactorial and possibly compounded in children with severe OSA.

### 3.4. Hypercoagulability

Disruption of the endothelium and exposure of subendothelial structure leads to activation of platelets and factors of the thrombolytic pathway and can result in a state of hypercoagulability. This may also contribute to atherogenesis. This has been reported in adult and pediatric patients with OSA [26].

### 3.5. Renal Injury and Microalbuminuria

Urine proteasomes were analyzed for unique proteins associated with OSA in a study involving 22 children with SDB. The premise is that abnormal metabolism from OSA will be measurable in the urine. The results demonstrated that there were elevated expression levels of gelsolin, perlecan, albumin, and immunoglobulin [29]. Gelsolin is commonly expressed in mammalian tissues and is implicated in the separation of actin filament. Perlecan is a glycosaminoglycan, a glomerular protein that is found in the urine associated with renal or urinary tract injury. This implies that there is a role in renal impairment in OSA, which may also contribute to hypertension resulting from OSA.

Microalbuminuria involves low-grade urinary protein loss. A pediatric study comparing children with OSA with controls (children with no OSA) showed that children with moderate to severe OSA are at higher risk of microalbuminuria, as evidenced by excretion of albumin in the morning urine. This may be due to the altered permeability in the glomerular system and may be related to hypoxemia during sleep, as well as OSA-induced activation of the sympathetic nervous system [30]. In a meta-analysis study of OSA and renal outcomes in adults, OSA was consistently associated with increased albuminuria or proteinuria and a decreased glomerular filtration rate [31].

### 3.6. Nonalcoholic Fatty Liver Disease

A recent Korean cross-sectional study showed that obese children with moderate and severe OSA had elevated alanine transaminase levels, which is one of the markers of nonalcoholic fatty liver disease in children (NAFLD) [32]. Another pediatric study exploring localized oxidative stress as a mediator for pediatric NAFLD found that nocturnal hypoxia, associated with OSA, is implicated in the progression of NAFLD, as well as nonalcoholic steatohepatitis (NASH) and liver fibrosis [33]. Evidence has shown that NAFLD in pediatric OSA populations that have undergone CPAP treatment appears to be improved, further demonstrating a link between NAFLD and OSA, while proposing a potential treatment for OSA-induced NAFLD in pediatric patients [34]. Adult OSA studies also showed that OSA is associated with the emergence of NAFLD, independent of obesity [35,36].

### 3.7. Gut Microbiota

OSA in children also alters the gut epithelium, leading to inflammation and translocation of bacteria to the gut epithelium [37]. One of the studies in animal models of OSA by Xu et al. reported the role of alterations in gut microbiota leading to the development of atherosclerosis. However, there was a lack of clinical studies looking at the interplay between OSA, cardiovascular disease, and microbiota. Further literature is warranted to explore the intermediate mechanisms between OSA, cardiovascular morbidities, and diseases that arise secondary to abnormalities of the gut, including irritable bowel syndrome (IBS) [38] and gastroesophageal reflux disease (GERD) [39], which are both associated with OSA.

## 4. Metabolic and Endocrine Dysfunction and OSA

Evidence suggests that untreated OSA increases the risk and incidence of insulin resistance, dyslipidemia, growth hormone dysfunction, and other metabolic disorders in children [40]. The bidirectional association between OSA and obesity plays a role in the development of metabolic and endocrine syndromes [40]. OSA also leads to alterations in the hypothalamic–pituitary–adrenal axis and irregular sympathetic activation [41]. A comprehensive overview of the metabolic alterations of OSA is depicted in Figure 2.

### 4.1. Insulin Resistance

There is a reported association of OSA with increasing insulin resistance when adjusted for confounding factors such as BMI and waist circumference [42]. Studies have even found an independent association between OSA and high fasting glucose and fasting insulin levels [43]. OSA leads to the development of type 2 diabetes mellitus (T2DM), which is characterized by insulin resistance in both adults and children [12,44]. Appropriate treatment of OSA leads to improvement in insulin resistance, as demonstrated in pediatric studies [45].

### 4.2. Inflammatory Markers

It should also be noted that inflammation mediates the interplay between OSA, obesity, and metabolic dysregulation. Studies have shown that certain inflammatory markers are elevated in obese pediatric patients [46]. These markers include interleukin (IL) 6, IL-8, IL-10, IL-17, IL-18, IL-23, macrophage migration inhibitory factor (MIF), highly sensitive C-reactive protein (Hs CRP), tumor necrosis factor-alpha (TNF-α), plasminogen activator inhibitor-1 (PAI-1), and leptin. Strong positive correlations were found between these markers and fasting insulin levels, BMI, apnea–hypopnea index (AHI), and other metabolic parameters [46]. These biomarkers also are implicated in the developments of cardiovascular morbidities. Prior studies have also identified IL-17 and IL-23 as being significantly increased in children with OSA, suggesting that they may serve as diagnostic markers for pediatric OSA [47].

### 4.3. Metabolic Syndrome

The complex interplay of OSA with obesity, hypertension, dyslipidemia, and insulin resistance can contribute to metabolic syndrome and can result in elevated levels of serum insulin, increasedblood pressure, triglycerides, and lower levels of high-density lipoproteins levels [12]. Metabolic syndrome itself is a risk factor for cardiovascular disease. Mechanisms of the development of OSA-induced metabolic syndrome include increased sympathetic activity, increased serum cortisol secondary to activation of the hypothalamic–pituitary–adrenocortical axis, and reactive oxidative species (ROS) formation [48]. Obesity is likely the primary mediator between OSA and metabolic syndrome [12,49]. C-reactive protein, adiposity, leptin, and insulin resistance were found to be associated with SDB in children, even after controlling for various demographic confounders [50,51].

### 4.4. Growth Failure

Studies have shown that growth is affected by OSA. Older studies have established a link between failure to thrive (FTT) and pediatric OSA, as demonstrated by an improvement in weight gain following AT [52] For this reason, FTT and associated impairments in growth have been implicated in OSA diagnosis. A pediatric study demonstrated that biomarkers of growth, particularly insulin-like growth factor binding protein 3 (IGFBP-3), were low in children with OSA, suggesting that there was growth hormone secretion impairment [53]. The study also demonstrated that resolution of OSA via AT leads to improved growth, most notably in weight, marked by significant elevations in circulating insulin-like growth factor-I (IGF-I) and IGFBP-3 concentrations. These improvements were significantly greater in the OSA-treated group than in the primary snoring group [53].

### 4.5. Polycystic Ovarian Syndrome

There appears to be somewhat of a bidirectional association between OSA and polycystic ovarian syndrome (PCOS), as reported in literature. The underlying mechanisms of both PCOS and OSA include insulin resistance, changes in levels of circulating hormones, ROS formation, and sympathetic activation [54,55]. Studies have demonstrated the potential of CPAP therapy in the treatment of both OSA and PCOS phenotypes. Given that both OSA and PCOS are risk factors for worsening metabolic and cardiovascular changes, treatment of both can be preventative against further comorbidities. As the prevalence of women with both OSA and PCOS increases, further research on the association between OSA and possible onset of PCOS in adolescent population is warranted.

### 4.6. Testosterone Deficiency

Though pediatric studies exploring the association between OSA and testosterone are not well established, adult studies have found an association between OSA and decreased levels of testosterone [56]. This is due to suppression of the hypothalamic–pituitary axis. Sleep fragmentation, sleep deprivation, frequent nocturnal awakenings, oxygen desaturation, and worsening of AHI can all contribute to lower levels of testosterone in males with OSA [56]. In particular, a negative correlation between OSA severity and testosterone levels has been identified. Furthermore, a higher AHI score is associated with lower testosterone levels.

### 4.7. Thyroid Function

Literature on OSA in children has highlighted associations between SDB and thyroid function. Sleep fragmentation and sleep deprivation, which are features of OSA, have been associated with lower levels of circulating thyroid stimulating hormone. Persistent OSA or untreated OSA in childhood can lead to hypothyroidism through suppression of the hypothalamic pituitary axis [57]. This may be related to the idea that the thyroid gland becomes a watershed area with limited perfusion during hypoxemia secondary to OSA although this has not been confirmed.

## 5. Neurocognitive Abnormalities and OSA

### 5.1. Cortical Thinning

Untreated and unmanaged pediatric OSA has been linked with significant neurocognitive sequelae. The hypoxemia and hypercapnia [58] that results from airway obstruction during sleep leads to oxidative stress and neuronal injury within the brain, particularly the hippocampus and cerebral cortex [59]. Lesions within the frontal lobe white matter were found on MRI of the brain in children with OSA [60]. Analysis of the cortical thickness using T-1-weighted MRI images as well as volumetric reconstructions of the subcortical structures in children with OSA suggested generalized cortical thinning within the medial orbital sulci [61]. In addition, another study, albeit in adult patients with OSA, further localized the cortical thinning in OSA patients to the orbital gyri, dorsolateral or ventromedial prefrontal regions, pericentral gyri, cingulate, insula, inferior parietal lobule, uncus, and basolateral regions [62]. Through analysis with memory tests, a higher number of respiratory arousals were noted not only in relation to cortical thinning of the anterior cingulare and inferior parietal lobule, but also longer apnea duration was related to cortical thinning of the dorsolateral prefrontal regions, pericentral gyri, and insula [62]. Decreased activation in the inferior parietal lobe was also noted, and this finding appears to be associated with impaired sensory input and processing [63]. Executive function, attention, and memory and learning are neurocognitive functions affected by OSA changes in the brain in adults [64].

### 5.2. Problem Solving and Executive Function

OSA has been found to be associated with dysfunction in overall executive function in children. Executive function includes inhibition, problem solving, fluid reasoning, and mental flexibility [63]. Inhibition is an all-encompassing term referring to deterring an automatic response to a stimulus. In patients with OSA, impaired inhibition is associated with poor impulse control. In children with untreated OSA, impairment in skills such as reading comprehension and mathematics was also reported [64]. Fluid reasoning is the ability to analyze problems by taking information from established knowledge and forming new connections within the brain, and this is noted to be affected in children with untreated OSA. Mental flexibility is essential for the brain to shift from one cognitive strategy to another. Impairment of this function in untreated OSA leads to decreased mental sharpness. It is plausible that untreated OSA can result in deficits that could worsen in children through the critical developmental years and affect learning, long term academic potential and subsequent occupational abilities.

### 5.3. Attention

Impaired attention is found in OSA children. Studies assessing polysomnograms and MRIs suggest that attention and vigilance are prominent neurocognitive functions that are impaired in children with OSA [65]. There are three components of attention, each of which can be compromised in patients with OSA. These include sustained attention, selective attention, and divided attention [66]. Whether these are related to endothelial dysfunction, hypoxemia, or both is still unknown.

Sustained attention includes attention required over extended lengths of time. In children with OSA, issues with sustained attention required during school days while juggling multiple classes, subjects, and extracurricular activities have been reported. Many of these children are concurrently diagnosed with attention-deficit/hyperactivity disorder, which is a neurocognitive disorder associated with a spectrum of attention and/or behavioral issues [67]. In those cases of complex patients, distinguishing between attention issues caused by ADHD versus those caused by OSA are difficult to delineate. Adequate treatment of pediatric OSA with concurrent ADHD treatment leads to improvements in behavior and attention [64].

Selective attention refers to the ability to focus on one task for a period of time. The task of driving requires selective attention amongst other skills such as the brain receiving visuospatial information. Studies analyzing adults with OSA have shown an increased incidence in motor vehicle accidents in those affected with OSA [67]. Further studies on the adolescent population are needed, especially many of whom driving is a new skill.

Lastly, the third component to attention is divided attention. Divided attention refers to the ability to be attentive to more than one stimulus at a time. When divided attention is impaired, individuals become overwhelmed with the filtering and processing of information. In children with OSA, impairment in this skill was reported [68]. When these skills are impaired, academic and functional abilities in these children are compromised.

### 5.4. Memory

Memory is an important component of the learning process. Recall processes from memory allow the storage of material to be used later. In patients with OSA, particularly in children and adolescents who are in school, OSA may affect episodic memory. Episodic memory includes immediate and delayed recall of events and experiences [69]. Two subcategories of episodic memory include visual and verbal components, including recalling of images and factual details [69]. Pediatric patients with OSA are more likely to have issues with visual and verbal episodic memory in not only the short-term recall, but also the long-term recall as well. It is important to note that increased hypoxemia and hypercapnia, decreased sleep quality, and increased obesity with its consequences are all factors that can impede the ability to recall and store new and old memories in children. Aside from its impact on memory required for general academics, OSA also had an impact on overall intelligence. Sleep deprivation from poor quality of sleep from OSA may worsen these impairments. Sleep deprivation from societal expectations or comorbid insomnia may further potentiate the sequelae.

### 5.5. School Performance

Obstructive sleep apnea affects problem solving, attention, and memory, and this ultimately causes repercussions in school performance. The foundation of knowledge and skills learned through school relies on children and adolescents to not only have these important neurocognitive skills, but also to use these skills to the best of the students’ abilities. When these factors become compromised due to OSA, this leads to worsening of school performance.

Two treatment modalities of OSA and its subsequent effects on school performance have been reported. The first is tonsillectomy and adenoidectomy, which is the first line of treatment in the pediatric OSA population [70]. A meta-analysis conducted in 2017 evaluated neurocognitive effects in children with OSA post-tonsillectomy and -adenoidectomy [71]. Subsequent neuropsychological testing conducted within these studies found improvements in patients’ executive function, attention, and memory and learning [71]. This allowed the students to not only improve their academic performance and use of class-time effectively, but also to improve their relationships with their teachers. Another study assessed outcomes of tonsillectomies and adenoidectomies for twelve months in preschool-aged children. Results from this study, however, did not find any treatment-attributable improvement in neurocognitive function in these patients [72]. One limitation of generalizing this isolated study is that it only evaluated patients with mild OSA. The second treatment modality of OSA is CPAP [73]. A meta-analysis evaluated 19 studies assessing neurocognitive function before and after CPAP management in adult OSA patients. Compared to pre-treatment patients, post-treatment patients were found to have improvement in neurocognitive domains of fluid reasoning, updating, inhibition, generativity, and shifting [73]. Moreover, another adult study correlated improvements in aspects of neurocognitive performance, including verbal fluency and working memory following oral appliance of consistent CPAP therapy [74]. The data on the effects on neurocognitive improvements with current treatment modalities in children are still lacking and will require further studies.

## 6. Psychological Syndromes and OSA

Several psychosomatic syndromes have been implicated in children with OSA, as depicted in Table 1. The incidence of somatic syndromes such as depression, anxiety, and insomnia are well correlated with corresponding levels of sympathetic arousal [75]. In a previous study, the Body Sensation Questionnaire (BSQ), a 17-item questionnaire grading increasing levels of somatic arousal, was administered to 152 upper airway resistance syndrome patients in 150 adult OSA patients. Increasing scores on BSQ were correlated with increased somatic syndromes in general, particularly anxiety (*p* < 0.0001) and insomnia (*p* ≤ 0.0001) Recent studies have explored the mechanisms underlying the associations between pediatric OSA and psychosomatic syndromes, as well as the persistence of these syndromes. However, currently there is a lack of longitudinal data to confirm these relationships as these pediatric patients become older.

### 6.1. Anxiety

Similar to the neurocognitive sequelae secondary to untreated and inadequately managed OSA, there is a range of psychiatric sequelae due to alterations of the brain’s neurotransmitters. Sleep functions as a health-regulating measure to balance physical and emotional health. Thus, sleep serves to regulate an individual’s mood, wellness, and overall quality of life. Due to the underlying disturbance in sleep in OSA, there are psychiatric and psychological repercussions. Patients with OSA have been found to have lower levels of GABA and higher levels of glutamate in the insular cortex [76]. This has consequently been associated with increased propensity for mental health disorders in OSA patients. For example, some studies have found an association between the neurotransmitter changes in adolescent and adults occurring as a result of sleep apnea and increased propensity of affective disorders, including generalized anxiety disorder and other anxiety disorders, as well as major depressive disorder [66] A higher incidence of these psychiatric sequelae has been reported in female OSA patients than male OSA patients [66]. This is especially relevant in the adolescent patient population, as this is the recommended time to begin diagnostic screening for mental health disorders. Adolescent patients begin to have increased responsibilities, including academic and family responsibilities and planning for the future. In combination with the added pressures of transitioning to adulthood, as well as the effects of sleep apnea, these patients have an increased risk for mental health issues.

### 6.2. Depression

It is important to note that there is overlap with the presentation of psychiatric sequelae of OSA and that of major depressive disorder. Some of these clinical symptoms include passivity, slowed psychomotor function, loss of facial expression, and lack of initiation [77]. Furthermore, hypersomnia can be present in both OSA and major depressive disorder [78]. Thus, medical providers may have difficulty in terms of differentiating and delineating between the diagnoses and assessing adequate management of each individual diagnosis. However, an important distinction is that while major depressive disorder can cause patients to have hypersomnia, increased fatigue, and poor motivation, the presence of apneic episodes is unlikely to be present in major depressive disorder [79]. OSA would instead be a secondary diagnosis to these patients. A meta-analysis was conducted to assess the relationship between depressive symptoms and OSA in pediatric patients up to 18 years old. The results of this study suggested that depressive symptoms are higher among children with OSA. This study also showed that fewer depressive symptoms were reported after AT compared to pre-surgery [80]. Overall, this emphasizes the importance of properly treating pediatric OSA patients as well as screening for mental health disorders in order for patients to be evaluated properly and receive proper interventions by psychiatrists, psychologists, and other mental health services.

### 6.3. Behavior

Another psychiatric sequela of OSA includes its impact on the biopsychosocial components of behavior. Many factors contribute to behavior. Not only is behavior impacted by biological and chemical components but it is also driven by psychology and largely impacted by our relationships with others and the environment. This is particularly relevant in younger pediatric patients, in whom behavioral issues tend to be more of a concern due to immaturity of the prefrontal cortex and impulse control simply due to younger age. Studies have found that in pediatric OSA, behavioral issues can be more prevalent than daytime sleepiness, which is more prominent in adults [81]. Due to this brain immaturity, as well as the brain changes with OSA, this can potentially lead to more inattention, hyperactivity, and impulsivity in children’s behavior. Children particularly lack impulse control, which is a significant factor when maintaining habits of good behavior while suppressing habits of rebellion and disregard for authority. With concurrent obstructive sleep apnea, this impulse control is augmented.

Management with tonsillectomy and adenoidectomy in pediatric OSA patients has helped to curb and regulate patient behavior. Prospective studies have analyzed students with OSA and their behavior following surgical treatment. Because behavior and attention are subjective measures, prospective studies quantified these changes in OSA patients with measures such as Tests of Variables of Attention (TOVAS), Pediatric Sleep Questionnaires, and Parent-Rating Scales [82]. Overall, these studies showed improvements in these markers of behavior, including improved attention and lessened impulsivity and hyperactivity. In 2013, the Childhood Adenotonsillectomy Trial (CHAT) study conducted a multicenter, single-blind, randomized control trial comparing the outcome between early AT versus watchful waiting in 464 children ages 5 to 9 with PSG-confirmed OSA [83]. In this study, there were greater improvements in observed behavior on the Conners’ Rating Scale and decreased scores in the Pediatric Sleep Questionnaire Sleep-Related Disorder Scale (PSQ-SRBD) and Epworth Sleepiness Scale (ESS) in patients who underwent early AT as compared to children in the watchful waiting group. This study however did not show a significant increase in attention and executive function on the Developmental Neuropsychological Assessment (NEPSY) in children post-AT compared to watchful waiting [83]. Another study conducted in 2012 evaluated behavior improvements in preschool-aged patients who were one year post-AT and compared to healthy controls. Results showed fewer behavioral problems, as attention and receptive vocabulary were improved, along with resolution of the airway obstruction in these OSA patients via surgical intervention [84]. Other studies have assessed improvements in OSA patient behavior status post-tonsillectomy and-adenoidectomy by comparing two different methods of follow-up polysomnograms vs. parent-reported severity of sleep-disordered breathing when making predictions of changes in behavior [85]. Ultimately, this study concluded that predictions of neurobehavioral morbidity in patients with OSA are difficult to gauge the degree of improvement [85]. However, overall, addressing the hypersomnolence and fatigue secondary to obstruction via surgical intervention fosters improvements in pediatric patient behavior, as well as their relationships with others and the environment.

### 6.4. Mood and Personality

In conjunction with stress, obstructive sleep can detrimentally affect interpersonal relations, mood, and personality. Even OSA that is mild has the capacity to deter social well-being, as patients have increased internal stressors that impact their physical and emotional health, leading them to not have enough energy to foster healthy relationships with others [86]. By not having healthy social relationships, these patients then have issues with their mood, which ties with mood disorders, as well as their concepts of self in terms of personality [87]. All of these factors are especially crucial in pediatric patients, as their social well-being, mood, and personality are not only impacted by their environment externally, but also internally in terms of their lifestyles, especially with sleep issues such as OSA. Overall, the importance of adequate and early management of OSA in pediatric patients serves to foster good behavior, protect against mood disorders such as anxiety and depression, improves stress levels, and altogether improves psychological well-being. Table 1 shows an overview of the sequelae of untreated pediatric OSA.

## 7. Conclusions

Through discussion of the cardiovascular, endothelial, metabolic, endocrine, neurocognitive, and psychological sequelae of OSA, it is crucial to ensure adequate treatment and management of pediatric OSA patients. Many of these sequelae are interconnected and have compounding effects on the patient’s overall health. OSA should, therefore, be considered a disease with multi-systemic consequences. Much of the current research and literature undermines the biopsychosocial sequelae in pediatric OSA. Inadequate management may potentially lead to detrimental health consequences. However, the link between pediatric OSA and OSA in adults is not yet fully understood. Future studies are needed so that this progression may be fully understood. There are uncertainties surrounding the specific mechanisms and implications regarding pediatric versus adult OSA. Treatment of OSA in childhood can lead to beneficial outcomes but may also prevent long-term sequelae. Healthcare providers have a responsibility to take part in this preventative care of patients through early screening and timely referral. Early intervention with adequate treatment of pediatric OSA can not only improve the patient’s physical health, but also fosters patient wellness and overall quality of life.

## Figures and Tables

**Figure 1 children-09-00261-f001:**
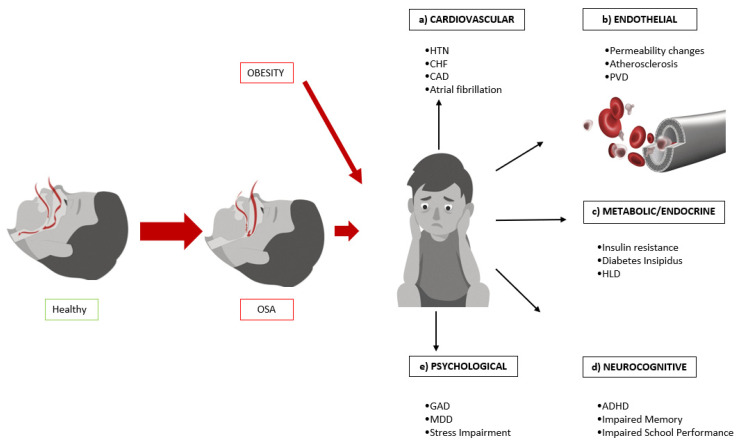
Sequelae of pediatric obstructive sleep apnea. (**a**) HTN = hypertension; CHF = congestive heart failure; CAD = coronary artery disease. (**b**) PVD = peripheral vascular disease. (**c**) HLD = hyperlipidemia. (**d**) ADHD = attention-deficit/hyperactivity disorder. (**e**) GAD = generalized anxiety disorder; MDD = major depressive disorder.

**Figure 2 children-09-00261-f002:**
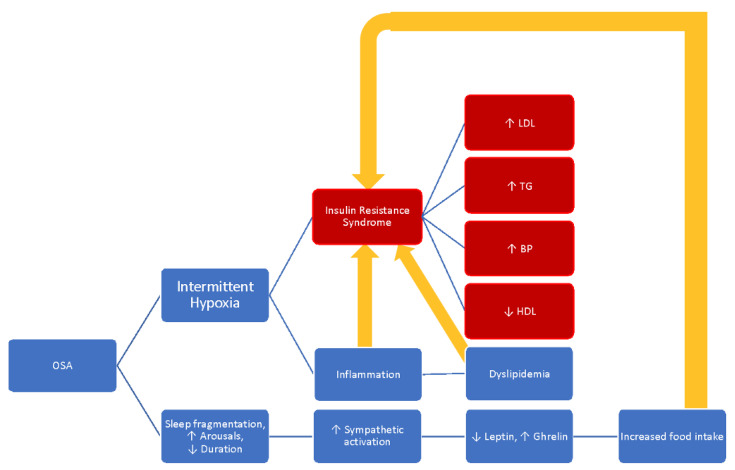
Metabolic alterations of obstructive sleep apnea. LDL = low-density lipoprotein; TG = triglyceride; BP = blood pressure; HDL = high-density lipoprotein.

**Table 1 children-09-00261-t001:** Sequelae linked to or complicated by poorly managed obstructive sleep apnea. The table lists a few affected systems in individuals with untreated OSA and specifies a few associated sequelae of the respective systems.

Affected System	Associated Sequelae
Cardiovascular	Hypertension, coronary artery disease, arrhythmias, cerebrovascular changes
Metabolic and endocrine	Insulin resistance, dyslipidemia, metabolic syndrome, growth changes, polycystic ovarian syndrome, testosterone deficiency, thyroid disorders
Endothelium	Vasoconstriction, atherosclerosis, hypercoagulability, renal damage
Gastrointestinal And Liver	liver disease, nonalcoholic fatty liver disease, irritable bowel syndrome, gastroesophageal reflux disease
Neurocognitive	Memory impairment, attention deficit hyperactivity disorder, executive function impairment
Psychosomatic	Anxiety, depression, behavioral issues

## Data Availability

Not applicable.

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
