# Peer review of "You Cannot Hit Snooze on OSA: Sequelae of Pediatric Obstructive Sleep Apnea"

_children, 2022, doi:10.3390/children9020261_

Round 1

Reviewer 1 Report

Interesting paper, some minor concerns to improve the overall quality:

Introduction

  • line 25, Pediatric OSA is mainly associated with adenoid and tonsils hypertrophy. Currently, adenotonsillectomy represents the treatment choice for OSA. Please cite doi:10.3390/children8100921.
  • line 120, Endothelial dysfunction of OSA patients has recently also been associated in subjects with vasomotor rhinitis, where vascular hyperactivity to chemical or thermal stimuli has been correlated with the severity of obstructive apneas. discuss and cite doi:10.3390/medicina56090454
  • line 220, Obstructive sleep apnoea (OSA) affects 11 percent of pre-menopausal women though it often remains undetected. Women may present differently than men, and the classic findings of snoring witnessed apnoeas and sleepiness may not be observed. Factors that predispose to OSA include polycystic ovarian syndrome, obesity, retromicrognathia, and hypothyroidism. OSA may contribute to neurocognitive dysfunction, depression, hypertension and metabolic syndrome. Emerging evidence indicates that snoring and OSA increase during pregnancy. For normal women with normotensive, low-risk pregnancies the prevalence of OSA is very low. Among normotensive pregnant women with high risk pregnancies, the prevalence of OSA is high and is even higher among those with gestational hypertension/preeclampsia during pregnancy. Incident snoring, which is a marker for OSA, is associated with an increased risk of developing gestational hypertension. please cite ''Champagne KA, Kimoff RJ, Barriga PC, Schwartzman K. Sleep disordered breathing in women of childbearing age & during pregnancy. Indian J Med Res. 2010;131:285-301.
  • line 248, Localized cortical thinning in OSA patients was found in the orbitorectal gyri, dorsolateral/ventromedial prefrontal regions, pericentral gyri, anterior cingulate, insula, inferior parietal lobule, uncus, and basolateral temporal regions at corrected P < 0.05. Patients with OSA showed impaired attention and learning difficulty in memory tests compared to healthy controls. A higher number of respiratory arousals was related to cortical thinning of the anterior cingulate and inferior parietal lobule. A significant correlation was observed between the longer apnea maximum duration and the cortical thinning of the dorsolateral prefrontal regions, pericentral gyri, and insula. please discuss and cite doi:10.5665/sleep.2876
  • line 312, neurocognitive performance improvement has been demonstrated in OSA patients treated with CPAP, oral appliance. Please discuss and cite doi:10.3390/bs11120180.

Author Response

Reviewer 1

Interesting paper, some minor concerns to improve the overall quality:

Introduction:

  • line 25, Pediatric OSA is mainly associated with adenoid and tonsils hypertrophy. Currently, adenotonsillectomy represents the treatment choice for OSA. Please cite doi:10.3390/children8100921.

Thank you for including this reference. Lines 25-27 have been modified to include the additional information and the citation has been included in the reference list.

  • line 120, Endothelial dysfunction of OSA patients has recently also been associated in subjects with vasomotor rhinitis, where vascular hyperactivity to chemical or thermal stimuli has been correlated with the severity of obstructive apneas. discuss and cite doi:10.3390/medicina56090454

Thank you for the suggestion and for including this reference. A few sentences have been added to acknowledge and include this adult study while calling for further research in pediatrics in lines 129-134. The citation has been added as well.

  • line 220, Obstructive sleep apnoea (OSA) affects 11 percent of pre-menopausal women though it often remains undetected. Women may present differently than men, and the classic findings of snoring witnessed apnoeas and sleepiness may not be observed. Factors that predispose to OSA include polycystic ovarian syndrome, obesity, retromicrognathia, and hypothyroidism. OSA may contribute to neurocognitive dysfunction, depression, hypertension and metabolic syndrome. Emerging evidence indicates that snoring and OSA increase during pregnancy. For normal women with normotensive, low-risk pregnancies the prevalence of OSA is very low. Among normotensive pregnant women with high risk pregnancies, the prevalence of OSA is high and is even higher among those with gestational hypertension/preeclampsia during pregnancy. Incident snoring, which is a marker for OSA, is associated with an increased risk of developing gestational hypertension. please cite ''Champagne KA, Kimoff RJ, Barriga PC, Schwartzman K. Sleep disordered breathing in women of childbearing age & during pregnancy. Indian J Med Res. 2010;131:285-301.

Thank you for the suggestion. This study presents an interesting subtopic that I would be interested in exploring further in a more comprehensive paper. Since it may be more appropriate to limit the paper to mostly pediatric studies, this citation may need to be omitted for the sake of organizational flow.  

  • line 248, Localized cortical thinning in OSA patients was found in the orbitorectal gyri, dorsolateral/ventromedial prefrontal regions, pericentral gyri, anterior cingulate, insula, inferior parietal lobule, uncus, and basolateral temporal regions at corrected P < 0.05. Patients with OSA showed impaired attention and learning difficulty in memory tests compared to healthy controls. A higher number of respiratory arousals was related to cortical thinning of the anterior cingulate and inferior parietal lobule. A significant correlation was observed between the longer apnea maximum duration and the cortical thinning of the dorsolateral prefrontal regions, pericentral gyri, and insula. please discuss and cite doi:10.5665/sleep.2876

Thank you for the suggestion. Please see lines 255-261 for further discussion and citations. Notation was made that this study was an adult study while the focus of the paper is with pediatric OSA patients.

  • line 312, neurocognitive performance improvement has been demonstrated in OSA patients treated with CPAP, oral appliance. Please discuss and cite doi:10.3390/bs11120180.

Thanks for the suggestion. Please see lines 337- 346 for discussion of neurocognitive improvement with CPAP.

Reviewer 2 Report

The authors present a nice far ranging review of sequelae of pediatric OSA. Overall, well-written and informative.

A few suggestions to consider for improvement:

  1. The authors should check there references for claims to ensure they are related to children versus adults. For example, the testosterone/OSA connection and reference 50- is this a study in children or adults? I think it is fine to include adult references, but it should be made clear in the text. Please check this issue throughout the manuscript.
  2. Regarding NAFLD and OSA, the authors may want to include studies examining this association and oxidative stress and changes with treatment. See: PMID: 27501738 & 29752170.
  3. See if there are any studies examining failure to thrive and OSA in kids to add to the "growth" section.

Author Response

Reviewer 2

The authors present a nice far ranging review of sequelae of pediatric OSA. Overall, well-written and informative.

A few suggestions to consider for improvement:

1) The authors should check there references for claims to ensure they are related to children versus adults. For example, the testosterone/OSA connection and reference 50- is this a study in children or adults? I think it is fine to include adult references, but it should be made clear in the text. Please check this issue throughout the manuscript.

Thank you for the suggestion. We have looked through the paper to find adult studies and have explicitly marked them in the manuscript as to avoid confusion. Thank you for pointing this out.

2) Regarding NAFLD and OSA, the authors may want to include studies examining this association and oxidative stress and changes with treatment. See: PMID: 27501738 & 29752170.

Thank you for the references. They both fit well in the discussion of NAFLD and OSA and help improve the organizational flow of the section. They have been added in the discussion in lines 169-175.

3) See if there are any studies examining failure to thrive and OSA in kids to add to the "growth" section.

Thank you for the suggestion. This has been elaborated upon in lines 226-229. Though the study happens to be older, it makes a good addition to that section.